# Explainable Spatio-Temporal Forecasting with Shape Functions

## Abstract

Spatio-temporal modeling and forecasting are challenging due to their complicated spatial dependence, temporal dynamics, and scenarios. Many statistical models, such as Spatial Auto-regression Model (SAR) and Spatial Dynamic Panel Data Model (SDPD), are restricted by a pre-specified spatial weight matrix and thus are limited to reflect its flexibility. Graph-based or convolution-based methods can learn more flexible representations, but they fail to show the exact interactions between locations due to the lack of explainability. This paper proposes a spatial regression model with shape functions to address the limitations of existing methods. Our method learns the shape functions by incorporating shape constraints, which are able to capture spatial variability or distance-based effects over distance. Therefore, our approach enjoys a learnable spatial weight matrix with a distance-based explanation. We demonstrate our method's efficiency and forecasting performance on synthetic and real data.

## 1 Introduction

Spatio-temporal data is widely observed in many areas, such as transportation (33; 27), climatology (2), and environmental research(19). The popularity of spatio-temporal data brings varieties of tasks for researchers, and one of the key tasks is forecasting. Spatio-temporal data has some inherent characteristics, namely, spatial dependence and temporal dynamics, which need to be considered for modeling and forecasting.

Spatial dependence means that the observations at different locations are not independent, and observations at closer locations often have a stronger correlation. In the statistics community, extensive research has been conducted to model spatial dependence, and various spatial models have been proposed. For example, in the spatial autoregressive (SAR) models, the spatial dependence is modeled by a product of an unknown parameter and a pre-specified spatial weight matrix (4; 1; 11; 12). Combined with the panel data, various types of spatial panel data models have been used to analyze spatio-temporal data (35; 13; 7; 22). One limitation of the autoregressive models is that the elements of the spatial weight matrix are pre-specified, such as an inverse distance. Although these pre-specified spatial weight matrices are applied to capture decreased distance-based effects, they fail to capture complex distance relations in real-world applications.

Researchers in the computer science community have developed various methods modeling spatio-temporal data using deep neural networks. Various neural network architectures have been proposed and applied to spatio-temporal forecasting, for example, spatio-temporal LSTM (31), fully connected gated graph architecture (20), Convolutional LSTM (23) and etc. One advantage of these methods is that they can incorporate unstructured data and rely on a high-performance computing platform to learn complicated representations for spatio-temporal problems. However, a critical limitation of these methods is that they fail to explain how the spatial interaction works explicitly. The lack of

Submitted to 36th Conference on Neural Information Processing Systems (NeurIPS 2022). Do not distribute.

interpretability restricts its reliability and deep insights into the underlying spatio-temporal process. The explanation can be obtained if we can estimate the coefficient matrix that intuitively explains spatio-temporal interactions.

In this paper, we propose an Explainable Spatio-Temporal Forecasting (ESTF) model, which utilizes a spatial autoregressive model with shape functions to address the current limitations. Our method extends the vector autoregressive (VAR) model (24) by incorporating distance information into the temporal coefficient matrix using shape functions (3). The shape constraints are designed to be consistent with the common fact that observations from neighbours have stronger spatial dependence versus long-distance pairs. It is known as Tobler's First Law, which is "Everything is related to everything else, but near things are more related than distant things"(26; 18). Unlike the pre-specified spatial weight matrix, this coefficient matrix is learnable and is thus more flexible in capturing real-world complex spatial relations. Moreover, the shape functions are represented as a combination of basis functions, and thus a smaller number of parameters needs to be estimated. Finally, ESTF can be easily extended to forecasting in non-stationary scenarios using a dynamic spatial weight matrix. We conduct experiments on both simulated and real data, and the results demonstrate that our method achieves better forecast accuracy and is computationally efficient and more explainable.

## 2   Related work

**Statistical models**    Several works focus on temporal dynamics when considering spatio-temporal forecasting problems. The classical time series models, such as VAR, and ARIMA models, are applied to spatio-temporal process modeling(21; 38). Besides, a spatial weight matrix is also introduced to the ARIMA model to capture spatial dependence (28). The non-stationarity, particularly unit-root non-stationarity, is mainly modeled by ARIMA or Co-integration models. In addition, spatial regression models or panel data are classical models in econometrics and can also be applied to model spatio-temporal problems. These models, for example, spatial auto-regression models, take spatial weight matrix into consideration and estimate parameters in the framework of regression. However, the common characteristics of these models need a pre-specified spatial weight matrix(35; 6). Elements in the matrices are generally an inverse distance of corresponding locations. Meanwhile, these spatial models focus on statistical inference on the scalar parameters placed before the spatial weight matrix(25). Although there are many choices for the spatial weight matrix, such as inverse distance, adjacency relationships, and K-nearest neighbors, there is a lack of research on estimating the spatial weight matrix. The pre-specified spatial weight matrix restricts models' application and fails to capture more complicated underlying spatial dependence. Some researchers developed a sparse spatio-temporal model that can estimate a sparse spatial weight matrix (17). The strict sparse setting also restricts the wide application of the spatial weight matrix.

**Graph-based methods**    Graph-based methods are widely applied for a non-Euclidean domain. Some types of spatio-temporal data, for example, traffic flow data or brain network data, can be represented as graphs. The graph structures well model the complicated spatial dependence. Thus, the definition or pre-specified graphs structure is normally required when developing a graph-based model. Related works can be found in (30; 14). The common typical method is GraphCNN, which is to apply a convolutional transformation to the neighbors of each node (29; 34). The graph convolution can capture patterns and features in the spatial domain. Graph-based methods have been proposed and widely applied to lots of real cases. Traffic flow data modeling and forecasting is a popular topic in this area (30; 20). Other topics, for example, climate sensor data (16), video (10) and etc, are also applied by variant graph-based models. RNN or LSTM combined with graphs, i.e., a sequence of graphs, are also considered in spatio-temporal forecasting problems (10).

**CNN-based methods**    Unlike graph-based methods, CNN-based methods are more suitable for modeling spatio-temporal data collected in regular grid locations. It applies filters to find relationships between neighboring inputs. Although some works (32) applied convolution neural networks to model non-grid traffic data, it is more common to see CNN-based methods process grid structures, e.g., images, video rather than a general domain. As some spatio-temporal data are collected from a regular grid in the Euclidean space (29), they thus can be viewed as a kind of special image. The CNN structure combined with RNN or LSTM has been developed to make forecasting for spatio-temporal data, for example, diffusion convolutional RNN (15), Convolutional LSTM networks (23; 36)and etc.

## 3 Proposed method

### 3.1 Problem formulation and notation

We use a $n \times 1$ vector $\mathbf{X_t} = \{\mathbf{x_{1t}}, \mathbf{x_{2t}}, \cdots, \mathbf{x_{nt}}\}$ to denote observations at time $t$, where $n$ is the number of locations. At each location $i$, $\mathbf{S_i} = (\mathbf{c_i^x}, \mathbf{c_i^y})$ is the coordinates of the location $i$. The distance between location $\mathbf{S_i}$ and $\mathbf{S_j}$ is $d_{ij} = \sqrt{(d_{ij}^x)^2 + (d_{ij}^y)^2}$, where $d_{ij}^x = |c_i^x - c_j^x|$ and $d_{ij}^y = |c_i^y - c_j^y|$. Our goal is to make forecasting for spatio-temporal data: given training data set $\mathbf{X_1}, \mathbf{X_2}, \cdots, \mathbf{X_T}$, we would like to make forecasting for the next $h$, $\hat{\mathbf{X}}_{T+1}, \cdots, \hat{\mathbf{X}}_{T+h}$.

### 3.2 The stationary spatio-temporal model with shape functions

We first consider the stationary case. To model the spatio-temporal stationary process, we consider the following model

$$\mathbf{X_t} = \sum_{\mathbf{k=1}}^{\mathbf{P}} \mathbf{W_k} \mathbf{X_{t-k}} + \epsilon_\mathbf{t}, \tag{1}$$

where $\mathbf{W_k}$ is a spatial weight matrix for capturing the spatial dependence at lag $k$, and $\epsilon_\mathbf{t}$ is white noise. Moreover, we assume the $(i, j)$th element of $\mathbf{W_k}$, $w_{ij}^{(k)}$, depends on the distance $d_{ij}$. That is, $w_{ij}^{(k)}$ depends on a function $f_k(d_{ij})$.

For spatio-temporal data, the spatial dependence, represented by $w_{ij}^{(k)}$, between locations decreases as the distance between two locations increases. In other words, there is a shape constraint for the function $f_k(d)$, such as a decreasing function. In order to estimate the shape function, we model $f_k(d)$ as a linear combination of basis functions $g_i(d)$, $i = 1, 2, \cdots, m$. More specifically, the shape function $f_k(d)$ is a linear combination of basis functions and coefficients with positive value $f_k(d) = a_{1,k}^2 g_1(d) + \cdots + a_{m,k}^2 g_m(d)$, where $a_{1,k}, \cdots, a_{m,k}$ are parameters to be estimated. The constraint of decrease needs parameters non-negative and thus each parameters squared. The spatial weight matrix can take the value of decreased shape function directly. The element of $\mathbf{W_k}$ is $w_{ij}^{(k)} = f_k(d_{ij})$. The details of the shape function and the corresponding basis functions can be found in Section 3.4

The parameters in shape functions can be estimated from the neural network illustrated in Figure 1. The neural network can be trained from the following criterion:

$$\min_{\{W_k\}_{k=1}^p} \sum_{t=1}^{T} ||\mathbf{X_t} - \hat{\mathbf{X}}_\mathbf{t}||^\mathbf{2} = \sum_{\mathbf{t=1}}^{\mathbf{T}} ||\mathbf{X_t} - \sum_{\mathbf{k=1}}^{\mathbf{P}} \hat{\mathbf{W}}_\mathbf{k} \hat{\mathbf{X}}_{\mathbf{t-k}}||^\mathbf{2}.$$

### 3.3 The non-stationary spatio-temporal model with time-variant shape functions

The static spatial weight matrix $\mathbf{W_k}$ can reflect spatial dependence and thus can be applied to stationary scenarios. Next, we consider the nonstationary case. Therefore, we extend the stationary model to non-stationary cases. The spatial weight matrices only reflect static relationships across time lags in the static model. Unlike these settings, we change spatial weight matrices to be time-variant. The spatial weight matrices formed by time-variant shape functions can thus capture non-stationary dynamic spatial dependence. The non-stationary model has the form below,

$$\mathbf{X_t} = \sum_{\mathbf{k=1}}^{\mathbf{P}} \mathbf{W_{t,k}} \mathbf{X_{t-k}} + \epsilon_\mathbf{t}. \tag{2}$$

where $\epsilon_\mathbf{t}$ is white noise, and $\mathbf{W_{t,k}}$ relies on shape function $f_{t,k}(d)$. Similar with stationary settings, the time-variant shape functions are still represented as a linear combination of basis functions $g_i(d)$, $i = 1, 2, \cdots, m$. The coefficients are therefore time-variant. The shape function at time $t$ has the form below $f_{t,k}(d) = a_{1,t,k}^2 g_1(d) + \cdots + a_{m,t,k}^2 g_m(d)$. Unlike stationary setting, the coefficients of nonstationary setting, $\{a_{i,t,k}\}_{i=1}^m$, depend on the time $t$. The non-stationary model can be trained from the criterion by minimizing

$$\min_{\{W_{t,k}\}_{k=1}^p} ||\mathbf{X_t} - \hat{\mathbf{X}}_\mathbf{t}||^\mathbf{2} = ||\mathbf{X_t} - \sum_{\mathbf{k=1}}^{\mathbf{P}} \hat{\mathbf{W}}_{\mathbf{t,k}} \hat{\mathbf{X}}_{\mathbf{t-k}}||^\mathbf{2}.$$

The networks for the stationary model as well as the non-stationary model are presented in the Figure 1.

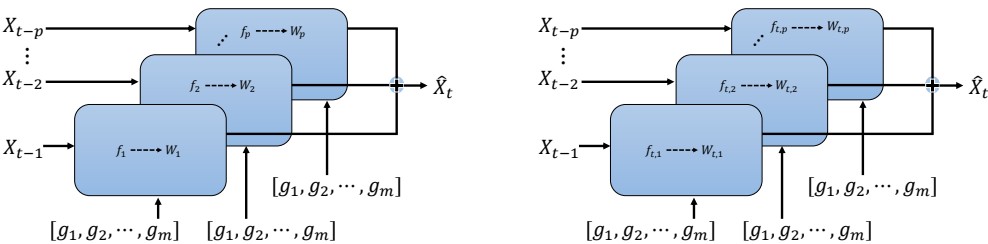

Figure 1: The neural network for the stationary spatio-temporal process (left) and non-stationary spatio-temporal process (right).

## 3.4 The basis functions for shape functions

The shape functions are integrated into our model to obtain distance-based explanations in stationary and non-stationary scenarios. The motivation of the proposed shape functions is that as the distance between two observations increases, the effects between these two locations decreases. These distance-based effects can be reflected in spatial weight matrix $\mathbf{W}$ and each element in the matrix can measure how the corresponding locations interact. The shape function is represented as a linear combination of basis functions. The basis functions, satisfying shape constraint, rely on the corresponding definition of basis functions.

**Definition of basis functions for various shape constraints.** We list the definition of basis functions for increased and decreased shape (3). The distance quantile among $\{d_{i_1,j_1}, d_{i_2,j_2}, , \cdots, d_{i_N,j_N}\}$ at quantile level $q_1, q_2, \cdots, q_m$ is denoted by $\{d_{(1)}, d_{(2)}, \cdots, d_{(m)}\}$, where $0 \leq q_1 < q_2 < \cdots < q_m \leq 1$ and $\{q_1, q_2, \cdots, q_m\} = \{\frac{1}{m}, \frac{2}{m}, \cdots, 1\}$. Here, we can set the number of $m << n^2$, and thus, the number of parameters is significantly reduced.

For the constraint of monotone decreasing function, the basis function is defined as $g_i(d) = \mathbf{1}_{\{\mathbf{d} < \mathbf{d}_{(i)}\}}$. The basis function for the shape function with the constraint of concave decrease is defined as $g_i(d) = (d_{(i)} - d)\mathbf{1}_{\{\mathbf{d}_{(i)} \leq \mathbf{d}\}}$ and convex decrease is defined as $g_i(d) = (d_{(i)} - d)\mathbf{1}_{\{\mathbf{d} \leq \mathbf{d}_{(i)}\}}$, for $1 \leq i \leq m$. Figure 2 shows the definition of basis functions for monotone decreased and increased shape functions, respectively. We only present four basis functions for each shape and each of them is related to four quantile levels. The dashed lines indicate the turning points for each basis function and they equal one or zero at the beginning and turn to zero or one at turning points.

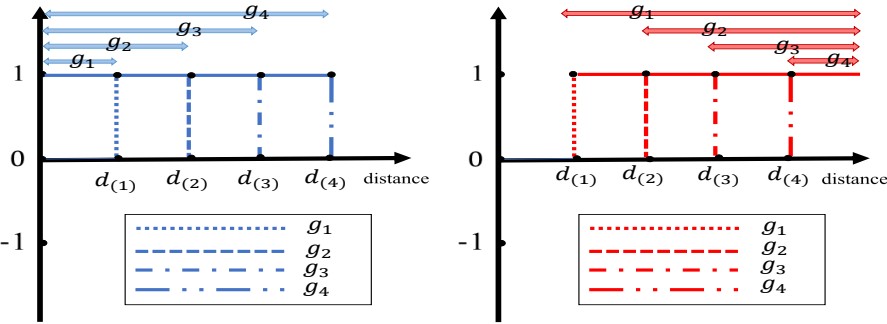

Figure 2: The basis functions for decreased shape (left) and for increased shape (right). The arrows indicate domain of each basis functions.

## 3.5 Model forecasting

The stationary model requires fixed shape functions and related spatial weight matrix are time-invariant. Given training data set $\mathbf{X_1}, \mathbf{X_2}, \cdots, \mathbf{X_T}$, we can estimate spatial weight matrix $\hat{W}_1, \hat{W}_2, \cdots, \hat{W}_p$ and make forecasting iteratively. That is $\hat{\mathbf{X}}_{T+1} = \sum_{k=1}^p \hat{W}_k \mathbf{X_{T+1-k}}$, $\hat{\mathbf{X}}_{T+2} = \hat{W}_1 \hat{\mathbf{X}}_{T+1} + \sum_{k=2}^p \hat{W}_k \mathbf{X_{T+2-k}}, \cdots \hat{\mathbf{X}}_{T+h} = \sum_{k=1}^p \hat{W}_k \hat{\mathbf{X}}_{T+h-k}$.

The non-stationary model incorporate time-variant spatial weight matrix $\hat{W}_{t,\cdot}$. Given the training data set $\mathbf{X_1}, \mathbf{X_2}, \cdots, \mathbf{X_T}$, we can obtain corresponding shape functions $\hat{f}_{1,\cdot}, \hat{f}_{2,\cdot}, \cdots, \hat{f}_{T,\cdot}$, where $\cdot$ denotes time lag. For lag $p = 1$, we can use $\{\hat{f}_t\}_{t=1}^T$ to represent time-variant shape functions for convenience. We can make dynamic forecasts for the next $h$ windows. One simple forecasting method is to use $\hat{W}_{T,k}$ to make forecast for $\hat{\mathbf{X}}_{T+h}$, that is

$$\hat{\mathbf{X}}_{T+h} = \sum_{k=1}^p \hat{W}_{T,k} \mathbf{X_{T+h-k}}.$$

The alternative method is to retrain the new forecast to obtain the latest shape functions as well as spatial weight matrix. Given long-term forecast window $L$, we first make short-term forecast for $h$ steps

$$\hat{\mathbf{X}}_{T+h} = \sum_{k=1}^p \hat{W}_{T+h,k} \mathbf{X_{T+h-k}},$$

where $h = 1, 2, \cdots$ and $\hat{W}_{T+h,k}$ is estimated by training forecast value of $\hat{\mathbf{X}}_{T+h-k}$. We repeat the process until $L$ steps in total have been predicted.

We summarize the whole process of our model when making spatio-temporal forecasts.

Step 1 Given the observation $\{\mathbf{X_t}\}_{\mathbf{t=1}}^{\mathbf{T}}$ and its coordinates, calculate all distance pairs among all locations, denoted by $\{d_{i_1,j_i}, \cdots, d_{i_N,j_N}\}$.

Step 2 Calculate $\{\frac{1}{m}, \frac{2}{m}, \cdots, 1\}$ quantile levels and obtain corresponding distance quantile value $\{d_{(1)}, d_{(2)}, \cdots, d_{(m)}\}$.

Step 3 Determine the shape constraints and construct corresponding basis functions. Specify the time lag $p$.

Step 4 Train the model according to the illustration of Figure 1.

## 4 Experiment

In order to assess our model in stationary and non-stationary scenarios, we synthesize data. Then, we apply our model to make some comparisons. On the one hand, we need to evaluate how the estimated shape functions look and assess their similarity and accuracy. On the other hand, our model can make spatio-temporal forecasting after estimating for spatial weight matrix. The basic idea for completing the two goals is to set up the expected shape function and compare estimated parameters with the real one. Next, we assess the forecasting performance with baseline models. Codes and data for replicating our experiments are anonymously published at `https://anonymous.4open.science/r/STVAR-F16E/`.

### 4.1 Simulation for stationary model

Here, we synthesize 100 stationary spatio-temporal data sets. The spatial domain consists of 30 locations and their coordinates can be found at `https://anonymous.4open.science/r/STVAR-F16E/`. For each location, we observe 500 values. The observation is generated from the stationary model $X_t = \sum_{k=1}^p W_k X_{t-k} + \epsilon_t$, where $\epsilon_t$ is randomly generated from the standard normal distribution. The next step is to construct random spatial weight matrices for each synthesized data set. The shape functions are set to be decreasing, and we set them as a logarithmic function:

$$\alpha(-\log(d+1) + \log(170)),$$

where $\alpha$ is randomly generated from uniform distribution [0.05,0.06] but kept to be fixed for each simulated data set. We use $d + 1$ to avoid zero value. This setting can make the real shape function

169  decrease and make it equal to zero when $d = 169$. The stationary model can iteratively generate the
170  $\mathbf{X_t}$ given initial value $\mathbf{X_0}$, where $\mathbf{X_0}$ is randomly generated from a uniform distribution with bounds
171  [-0.01,0.01]. The time lags are set as $p = 1$.

172  **Estimation for shape functions.** In Figure 3,
173  the estimated shape function is presented in red,
174  while the real shape function is presented in blue.
175  It can be seen that the estimated shape function
176  can capture the trend of the real shape function.

177  **Training details.** The first 300 steps are used as
178  training data, saving the last 200 steps for eval-
179  uation. We train all models for 100 epochs with
180  Adam optimizer (5) and a learning rate of 0.01.
181  The process involves parallel training across 10
182  CPUs. We select 100 quantile levels, and thus
183  100 basis functions $g_i(d)$ were generated as the
184  inputs for the model.

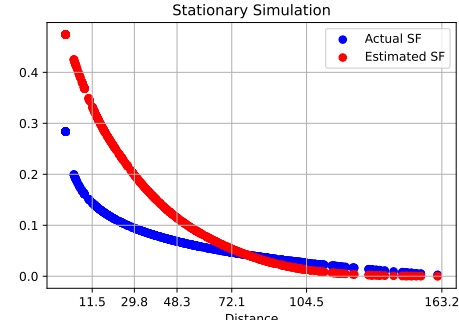

Figure 3: The sample of estimated shape function. Distances are shown every $20^{th}$ quantile.

185  **Assessment for forecasting.** We assess the fore-
186  casting performance for the stationary model
187  with baseline models. As introduced in the liter-
188  ature review, the baseline models are selected from the VAR model(21), the spatial panel data(SPE)
189  model that applied pre-specified spatial weigh matrix (28), graph-based models (20; 37) and
190  convolution-based models (15; 23). The error metrics are mean absolute error and root mean squared
191  error defined by $\frac{1}{Nn} \sum_{j=1}^{N} \sum_{i=1}^{n} \frac{\sum_{t=T}^{T+h} |\hat{X}_{it}^{(j)} - X_{it}^{(j)}|}{h}$, $\frac{1}{Nn} \sum_{j=1}^{N} \sum_{i=1}^{n} \sqrt{\frac{1}{h} \sum_{t=T}^{T+h} (X_{it}^{(j)} - \hat{X}_{it}^{(j)})^2}$,
192  respectively. The Table 1 shows the six baseline models with the proposed model. As totally we
193  have 100 synthesised data sets, $X_{it}^{(j)}$ and $\hat{X}_{it}^{(j)}$ denote $i - th$ variable in $j - th$ data sets. $n = 30$ is
194  the number of locations and $N = 100$ is the number of synthesised data. We conducted one-step
195  forecasting for the next 200 observations.

196  Compared with baseline models, the proposed model performs better under the metric MAE and
197  RMSE. The proposed method outperforms the closest competing method, DC-RNN, by 10%.

198  ## 4.2  Experiments for non-stationary model

We conduct a simulation for the non-stationary model with time lag $p = 1$ and synthesize 100 data
sets using a similar approach to the stationary model simulation. The initial value $\mathbf{X_0}$ and $\epsilon_t$ are
generated from a uniform and normal distribution respectively. The locations of observations are the
same as those in the stationary model simulation. In order to construct $W_t$, the time-varying shape
functions are created under the decreased constraint. The shape function at time $t$ is constructed as

$$\alpha_t(-\log(d + 1) + \log(170)),$$

199  where $\alpha_t$ controls the level of value at each time $t$. $\epsilon_t$ is generated from a normal distribution. $\mathbf{X_0}$ is
200  generated from a uniform distribution with bound [-0.001,0.001].

201  **Shape functions settings and estimation.** The shape functions are set as time-variant, as they can
202  simulate the non-stationary process across time. We specified $\alpha_0$ at $t = 0$ from uniform distribution
203  $[1 \times 10^{-4}, 2 \times 10^{-4}]$ and then make an interpolation from $\alpha_0$ to $\alpha_{500}$. The total length for every
204  location is 500 and we set $\alpha_{500} = 10 \times \alpha_0$. For example, generally if $\alpha_0 = 0.0001$, we have
205  $\alpha_t = 0.0001(1 - \frac{t}{T}) + 0.001\frac{t}{T}$, where $T = 500$. This setting guarantee that shape functions vary
206  from lower level to higher level. The larger $\alpha_t$ is, the more larger distance-based effects they have.
207  Thus, the corresponding spatial weight matrix consists of dynamic shape functions and can reflect the
208  non-stationary dependence among each site. We present the estimated shape functions in Figure 4
209  and compare them with the real ones.

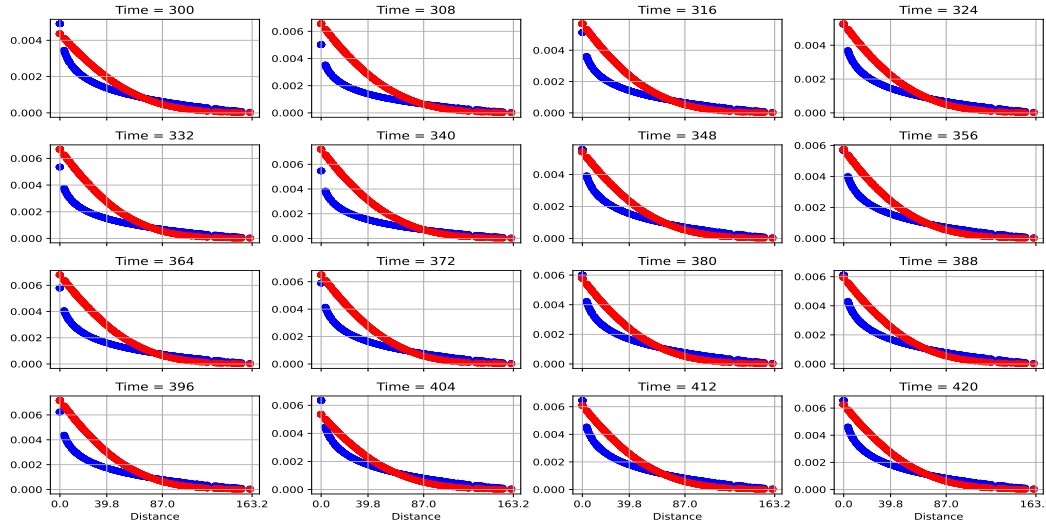

Figure 4: The sample of estimated shape function for the 120 testing time steps. Distances are shown every $40^{th}$ quantile.

**Training details.** Similar to the stationary simulation, the train-test split is $300 - 200$ over the data size of $500$. However, we train all models for $100$ epochs with Adam optimizer (5) at a learning rate of $0.001$. We train models in parallel across 10 CPUs.

**Forecasting performance.** The forecasting performance is assessed by the same metrics used in the previous simulation for the stationary case. We made a one-step forecast by our model. As for the baseline models, we adjusted their published code accordingly. The results show that the proposed model can still capture non-stationary processes compared with baseline models. The proposed method outperforms the other competing methods. The error metric is shown in Table 1.

Table 1: The error metrics with baseline models for simulation.

| Methods | Stationary Simulation | | Non-stationary Simulation | |
|---|---|---|---|---|
| | MAE | RMSE | MAE | RMSE |
| VAR | $2.9611 \pm 1.8573$ | $3.2588 \pm 1.8077$ | $2.4426 \pm 1.2285$ | $2.7676 \pm 1.2015$ |
| SPM | $1.8850 \pm 0.6348$ | $1.8671 \pm 0.6778$ | $2.1918 \pm 0.7350$ | $2.2161 \pm 0.6876$ |
| DC-RNN | $0.8960 \pm 0.0370$ | $1.1168 \pm 0.0426$ | $0.9017 \pm 0.0358$ | $1.1328 \pm 0.0463$ |
| FC-GAGA | $2.5425 \pm 0.2965$ | $3.1066 \pm 0.3633$ | $1.0270 \pm 0.0080$ | $1.2939 \pm 0.0120$ |
| GMAN | $1.6806 \pm 0.1491$ | $1.9293 \pm 0.1483$ | $1.5714 \pm 0.1104$ | $1.8608 \pm 0.1155$ |
| ConvLSTM | $2.9495 \pm 0.2980$ | $3.2509 \pm 0.2887$ | $2.2478 \pm 0.2295$ | $2.5469 \pm 0.2324$ |
| **ESTF** | $\mathbf{0.7997 \pm 0.0015}$ | $\mathbf{1.0017 \pm 0.0016}$ | $\mathbf{0.8075 \pm 0.0016}$ | $\mathbf{1.0112 \pm 0.0020}$ |

### 4.3 Real case studies

**Air quality data.** We apply our model to air quality data, which records air quality in California over 2021 [1]. The daily mean of PM 2.5 is recorded across 172 sites.

We obtain the first 200 steps for training and perform forecasting for the next 165 steps. All models are trained for 100 epochs using Adam optimizer (5), at a learning rate of 0.01 and batch size of 50. We present the estimated time-variant shape functions in supplemental file. The value of shape functions decays to zero at around 5.926, which is 80% quantile in the sample of distance pairs. In other words, the distance-based effects decay to zero at a distance equal or larger than 5.926. Our

---

[1]https://www.epa.gov/outdoor-air-quality-data/download-daily-data

model has ideal performance with low time consummation compared with baseline models. We put
detailed forecasting results of simulation and real cases in a supplemental file.

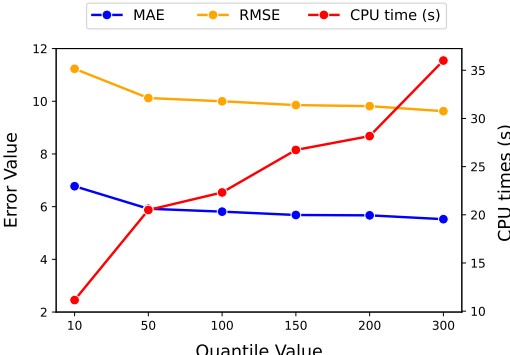

Figure 5: Comparing efficiency vs. performance trade-off at different quantile values.

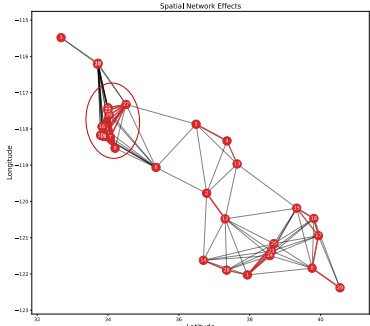

Figure 6: The significant distance-based effect among all 30 locations.

The result is shown in Table 2. The ESTF performs best in terms of RMSE, while the DC-RNN
method performs best in terms of MAE. For the computational time, the ESTF method is significantly
faster than most machine learning methods, and only takes around $1/10$ time of DC-RNN. In Figure 5,
MAE, RMSE, and time are presented with different numbers of $m$. As $m$ increases, the computational
time increases while both MAE and RMSE decrease. There is a significant increase in the forecasting
performance when $m$ increases from 10 to 50. For $m > 50$, the forecasting performance does not
increase much as $m$ increases.

One key advantage of the ESTF method is that we can make an explicit distance-based explanation
for our dataset. Figure 6 shows the distance-based effects at time $t = 9$. We only present the effects
using a threshold to obtain a more concise visualization. The estimated shape function $\hat{f}_9$ ranges
from 0 to 9.8 and we set 5 as the threshold. The red line indicates the value of the shape function
larger than 7, while the gray line indicates the value between 5 and 7. Figure 6 shows how any two
locations interact and measure the distance-based effects quantitatively. For example, air quality
monitoring sites around the Greater Los Angeles(red circle in Figure 6) area have a strong spatial
interaction with each other, such as node 7 and node 8.

### 4.4 $SO_2$ **data**

Texas is the second largest manufacturing state in the USA and prediction for $SO_2$ is critical task for
researchers. The data [2] records daily $SO_2$ at 31 locations in 2021. More detailed spatial information
can be found in the supplemental file. The numeric result is listed in Table 2.

Table 2: The error metrics with baseline models for real case study. Clock time (in seconds) for real
case study is recorded when training each model for 100 epochs on a single CPU.

| Methods | | | Air quality data | | | | $SO_2$ data | |
|---|---|---|---|---|---|---|---|---|
| | MAE | RMSE | Training time (s) | Inference Time (s) | MAE | RMSE | Training Time (s) | Inference Time (s) |
| VAR | 16.9844 | 22.3410 | 3.56 | 0.04 | 6.2705 | 9.1388 | 3.330 | 0.016 |
| SPM | 8.4547 | 13.8262 | 0.31 | 0.03 | 7.1453 | 9.1086 | 0.143 | 0.027 |
| DC-RNN | **4.7157** | 9.3873 | 203 | **1.211** | **3.5094** | 6.8681 | 264.215 | **1.366** |
| FC-GAGA | 7.8671 | 18.1870 | 181 | 2.759 | 4.5976 | 7.7528 | 169.425 | 2.889 |
| GMAN | 12.5268 | 17.3817 | 140 | 1.823 | 4.1099 | 7.4806 | 172.016 | 1.581 |
| ConvLSTM | 12.6292 | 17.9149 | 53 | 1.940 | 4.1445 | 8.0688 | 96.233 | 1.656 |
| **ESTF** | 5.2237 | **9.2169** | **22** | 1.625 | 4.2966 | **6.8307** | **31.050** | 1.868 |

Similar conclusions can be drawn in $SO_2$ data as that of air quality data. The ESTF model performs
best under the RMSE metric, while DC-RNN is best in the MAE metric. In terms of training time,

---

[2]https://www.epa.gov/outdoor-air-quality-data/download-daily-data

the proposed ESTF method costs around 10% of that of DC-RNN. For the inference time, ESTF and DC-RNN are comparable.

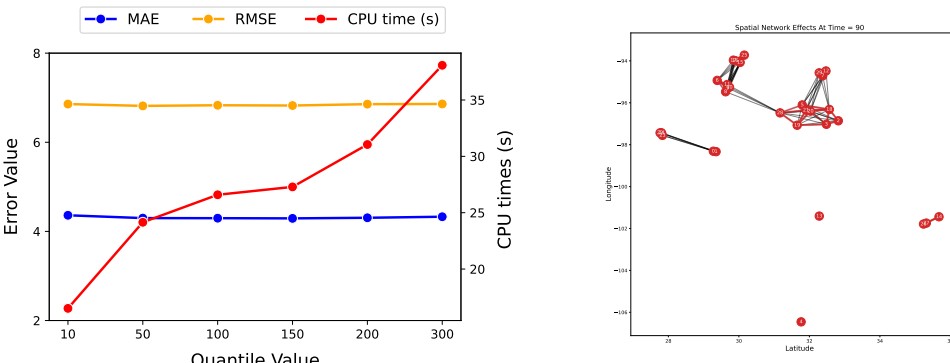

Figure 7: Comparing efficiency vs. performance trade-off at different quantile values.

Figure 8: The significant distance-based effect among all 31 locations at $t = 90$

The efficiency analysis and performance at different quantiles are shown in Figure 7. Together with Figure 5, we can see that the increasing number of basis functions does not have much improvement when the number of basis functions is larger than 50, while the training time increases as the number of basis functions increases. The spatial distribution at time $t = 90$ is presented in Figure 8 where coordinates are denoted by latitude and longitude. Two significant clusters, representing Houston and Dallas respectively, have the strongest distance-based effect. It quantitatively shows how these neighbors affect each other. Counties around Dallas-Fort Worth metropolitan area show strong interaction, which should be noted by environmental policy-makers. More detailed results are presented in the supplemental file.

## 5    Discussion

This paper applies learnable shape functions to capture distance-based effects. It can model dynamic spatial dependence for stationary and non-stationary spatio-temporal data based on their distance. The model does not have the limitations of classical statistical spatial models and provides a more explanatory model than usual deep learning methods. Furthermore, some spatio-temporal data, such as temperature for sea surface and air quality monitoring data, usually viewed as collected from the continuous field, are more suitable for the proposed models since these kinds of data follow the basic rule that variability between two locations is significantly affected by their distance. However, some spatio-temporal data, such as traffic flow or some biology data, do not follow the rule. As a result, the spatial dependence may rely on road structure or biological mechanisms instead of distance. It is worth researching such data by considering graph structure when estimating spatial weight matrix. In addition, we can develop spatio-temporal causal inference based on the ESTF model. Grander causal analysis can be done by fitting the first-order VAR model (24). The estimation of the coefficients matrix of the VAR model attracts researchers' interest as it can be treated as a causal transition matrix. In the causal inference community, lots of work have been conducted on the VAR model (8; 9). However, there is a lack of research on causal inference under the spatio-temporal process. The quantitative distance-based effects in ESTF can be further researched and extended to develop a spatio-temporal causal model.

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
