# Supplemental material for: Explainable Spatio-Temporal Forecasting with Shape Functions

## A    Simulation for stationary model

We present additional simulation results of the synthetic stationary data in the section. Figure 1 shows the one simulated dataset, where the blue line indicates the testing data from time $t = 300$ to $t = 500$, and the red line shows the forecasting line. The spatial domain consists of 30 locations that have corresponding X and Y coordinates, presented in Table 1. Figure 1 and Figure 2 show that the ESTF model can accurately forecast the trend of stationary spatio-temporal processes.

Table 1: The spatial domain in simulation study

| Data Index | X Coordinate | Y Coordinate | Data Index | X Coordinate | Y Coordinate | Data Index | X Coordinate | Y Coordinate |
|---|---|---|---|---|---|---|---|---|
| **1** | 63 | 10 | **2** | 95 | 30 | **3** | 96 | 16 |
| **4** | 37 | 36 | **5** | 55 | 16 | **6** | 143 | 28 |
| **7** | 12 | 18 | **8** | 25 | 32 | **9** | 41 | 18 |
| **10** | 67 | 24 | **11** | 173 | 28 | **12** | 112 | 20 |
| **13** | 59 | 10 | **14** | 99 | 18 | **15** | 128 | 30 |
| **16** | 10 | 21 | **17** | 106 | 22 | **18** | 110 | 10 |
| **19** | 105 | 30 | **20** | 98 | 10 | **21** | 111 | 6 |
| **22** | 120 | 20 | **23** | 55 | 12 | **24** | 179 | 28 |
| **25** | 99 | 26 | **26** | 27 | 24 | **27** | 109 | 32 |
| **28** | 65 | 6 | **29** | 159 | 8 | **30** | 9 | 14 |

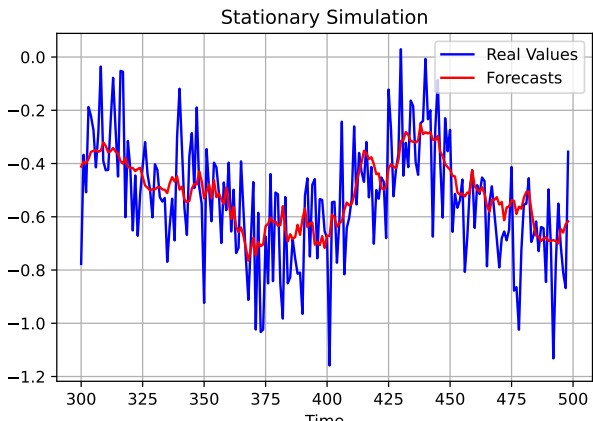

Figure 1: The sample of forecasting results at one location in the stationary case.

Submitted to 36th Conference on Neural Information Processing Systems (NeurIPS 2022). Do not distribute.

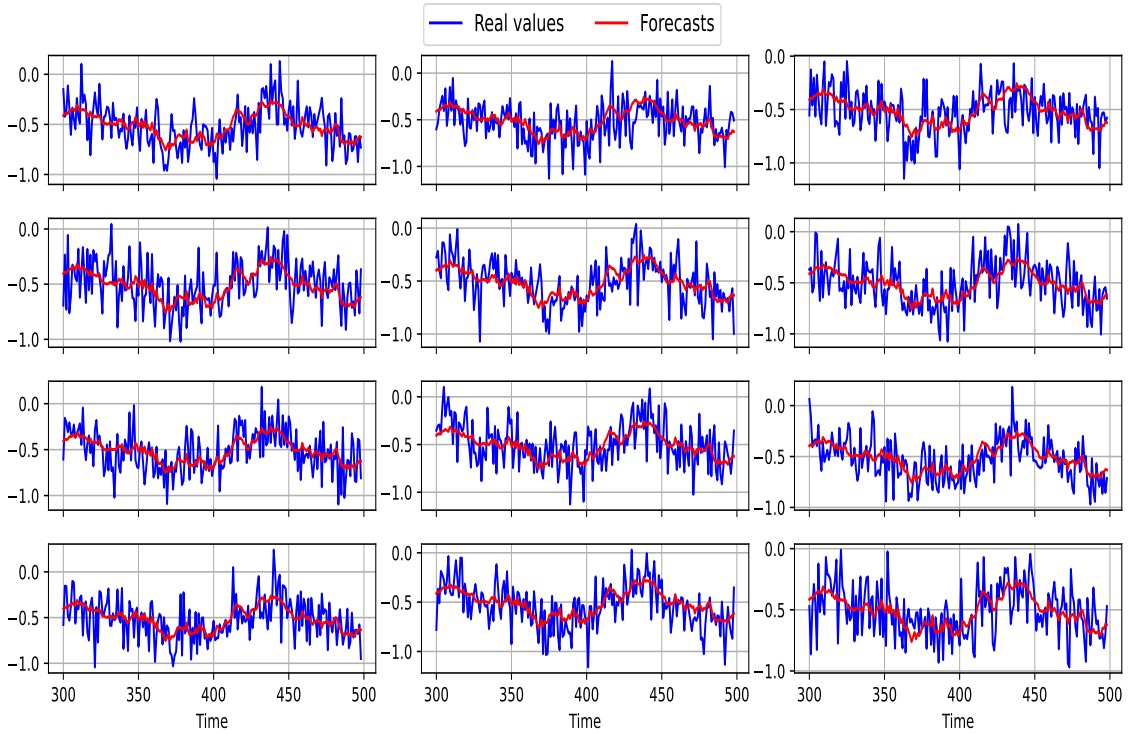

Figure 2: The sample of forecasting results in other random 12 locations in the stationary case.

## 7 B Experiments for non-stationary model

8 The simulation results for non-stationary simulation are presented in Figure 3 and Figure 4, where
9 the blue line indicates the testing data from time $t = 450$ to $t = 500$, and the red line shows the
10 forecasting line. The spatial domain is the same as the one in the stationary simulation study. As
11 shown in Figure 3 and Figure 4, the ESTF model can capture the trend accurately.

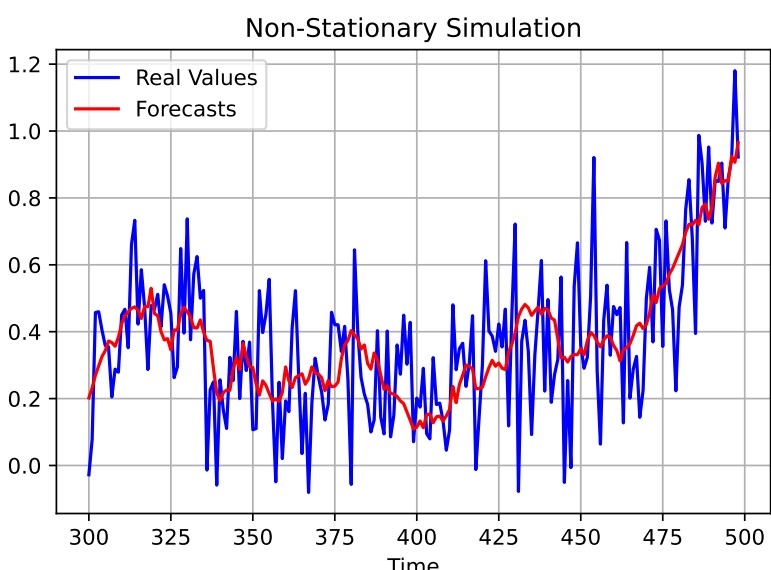

Figure 3: The sample of forecasting results in one location for the non-stationary case.

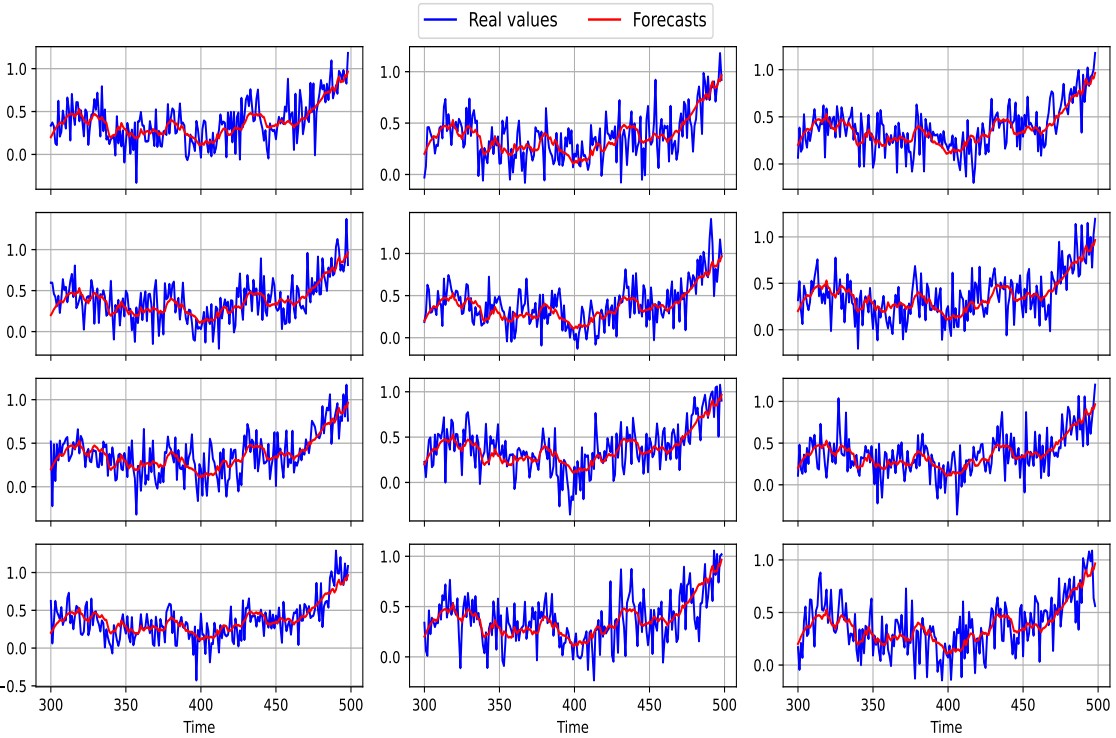

Figure 4: The sample of forecasting results for 12 random locations for the non-statioanry case.

## 12 C    Real case study

### 13 C.1    Air quality data

14 We present the forecasting performance of the ESTF model and spatial distance-based effects in
15 the section. The spatial information of 30 locations is presented in Table 2. To train the ESTF
16 model, we apply the first 200 data as training data and make a one-step forecast for the next 165 time
17 points. Figure 5 and Figure 6 show the forecasting performance at one location and other 12 random
18 locations, respectively. The model can also make accurate forecasts for the trend and daily variability.
19 The spatial network at different times is also presented in Figure 8 to Figure 11. Red lines show the
20 connection larger than 75% of maximum value of estimated shape functions while gray lines are
21 those between 50% and 75%. Locations around the Greater Los Angles, centered around 34 latitude
22 and -118 longitude, exhibit strong distance-based effects among each other.

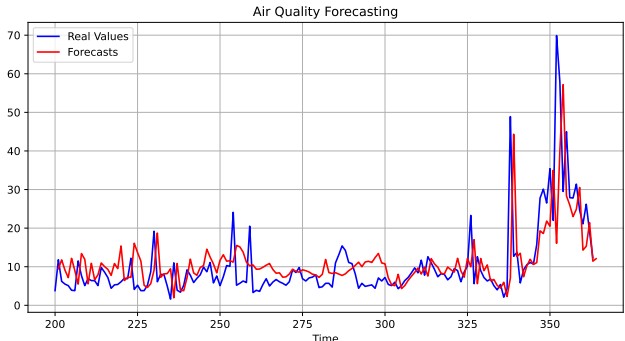

Figure 5: The sample of forecasting of air quality in one location for the next 165 days.

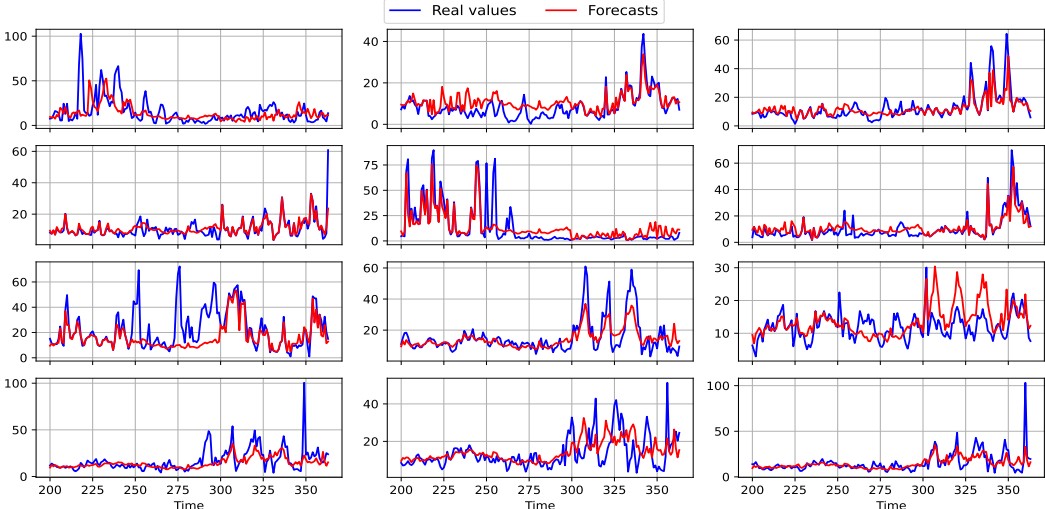

Figure 6: The sample of forecasts for other 12 locations in the real case study.

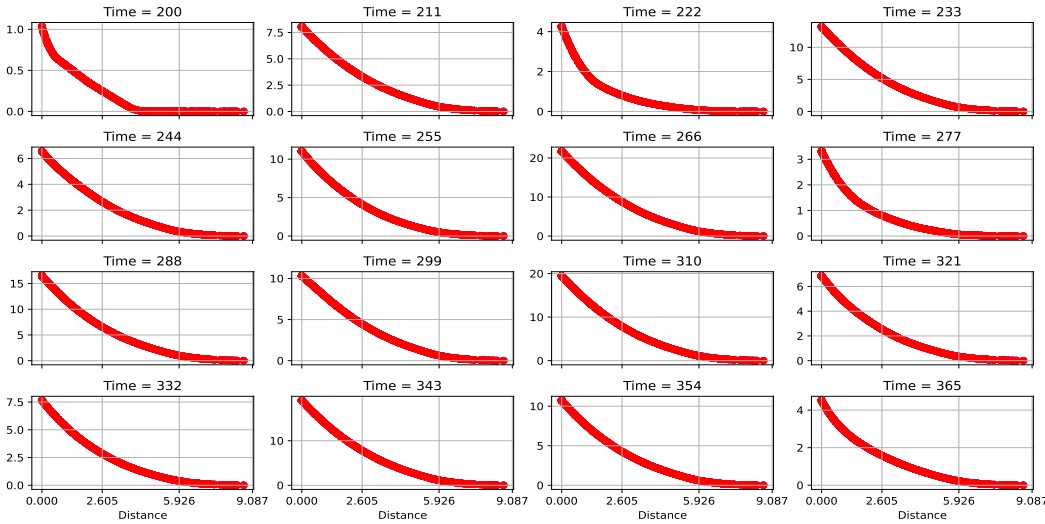

Figure 7: The samples of estimated shape function within the next 165 testing time steps. Distances are shown every $40^{th}$ quantile.

The code can be accessed from https://anonymous.4open.science/r/STVAR-F16E/. Table 2 shows the spatial information for the 30 locations in the real case study, which include the site name, county code, state code, latitude, and longitude.

Table 2: The sites information for Air quality data at 30 locations .

| Data Index | Site Name | County Code | State Code | latitude | longitude |
|---|---|---|---|---|---|
| 0 | Chico-East Avenue | 7 | 6 | 39.76168 | -121.84047 |
| 1 | Concord | 13 | 6 | 37.936013 | -122.026154 |
| 2 | Fresno - Garland | 19 | 6 | 36.78538 | -119.77321 |
| 3 | Calexico-Ethel Street | 25 | 6 | 32.67618 | -115.48307 |
| 4 | White Mountain Research Center | 27 | 6 | 37.360684 | -118.330783 |
| 5 | Keeler | 27 | 6 | 36.487823 | -117.871036 |
| 6 | Bakersfield-California | 29 | 6 | 35.356615 | -119.062613 |
| 7 | Los Angeles-North Main Street | 37 | 6 | 34.06659 | -118.22688 |
| 8 | Reseda | 37 | 6 | 34.19925 | -118.53276 |
| 9 | Compton | 37 | 6 | 33.901389 | -118.205 |
| 10 | Long Beach (South) | 37 | 6 | 33.79236 | -118.17533 |
| 11 | Long Beach-Route | 37 | 6 | 33.859662 | -118.200707 |
| 12 | Merced-M St | 47 | 6 | 37.30832 | -120.480456 |
| 13 | Mammoth | 51 | 6 | 37.64571 | -118.96652 |
| 14 | Salinas 3 | 53 | 6 | 36.694261 | -121.623271 |
| 15 | Truckee-Fire Station | 57 | 6 | 39.32783 | -120.184592 |
| 16 | Anaheim | 59 | 6 | 33.83062 | -117.93845 |
| 17 | Quincy-N Church Street | 63 | 6 | 39.939567 | -120.944376 |
| 18 | Portola | 63 | 6 | 39.81336 | -120.47069 |
| 19 | 29 Palms | 65 | 6 | 33.71969 | -116.1897 |
| 20 | Indio | 65 | 6 | 33.70853 | -116.21537 |
| 21 | Rubidoux | 65 | 6 | 33.99958 | -117.41601 |
| 22 | Mira Loma (Van Buren) | 65 | 6 | 33.99636 | -117.4924 |
| 23 | Sacramento-Del Paso Manor | 67 | 6 | 38.613779 | -121.368014 |
| 24 | Sacramento-1309 T Street | 67 | 6 | 38.56844 | -121.49311 |
| 25 | Folsom-Natoma St | 67 | 6 | 38.683304 | -121.164457 |
| 26 | Ontario-Route 60 Near Road | 71 | 6 | 34.030833 | -117.61722 |
| 27 | Victorville-Park Avenue | 71 | 6 | 34.51096111 | -117.32554 |
| 28 | San Jose - Jackson | 85 | 6 | 37.348497 | -121.894898 |
| 29 | Redding - Health Department | 89 | 6 | 40.55013 | -122.38092 |

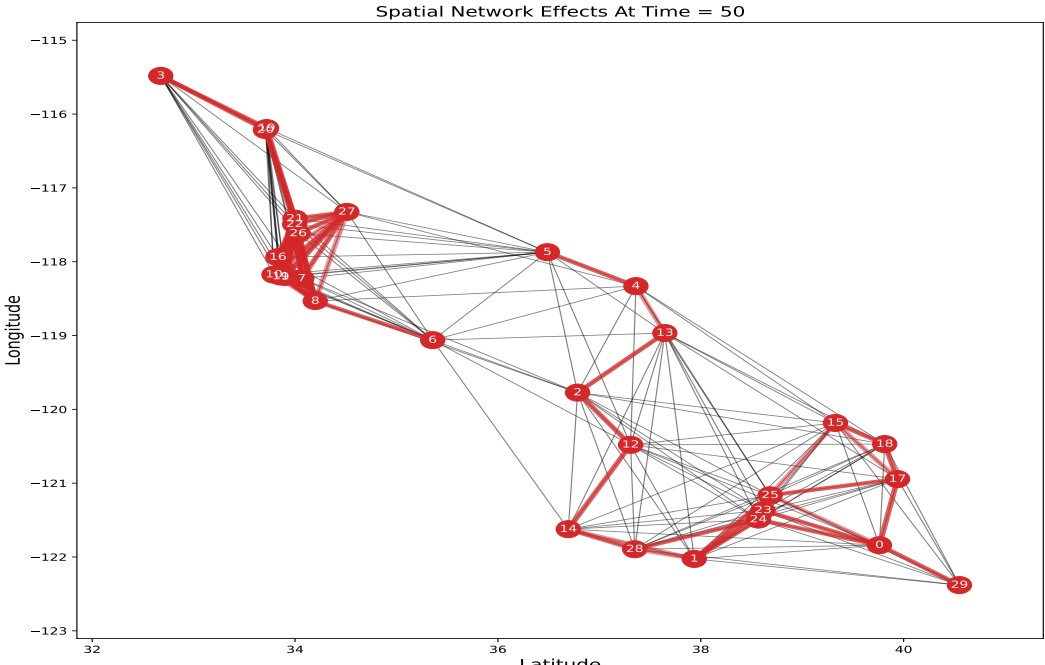

Figure 8: The distance-based effect among all 30 locations at time $t = 50$. Red lines show the connection larger than 75% of maximum value of estimated shape functions while gray lines are those between 50% and 75%.

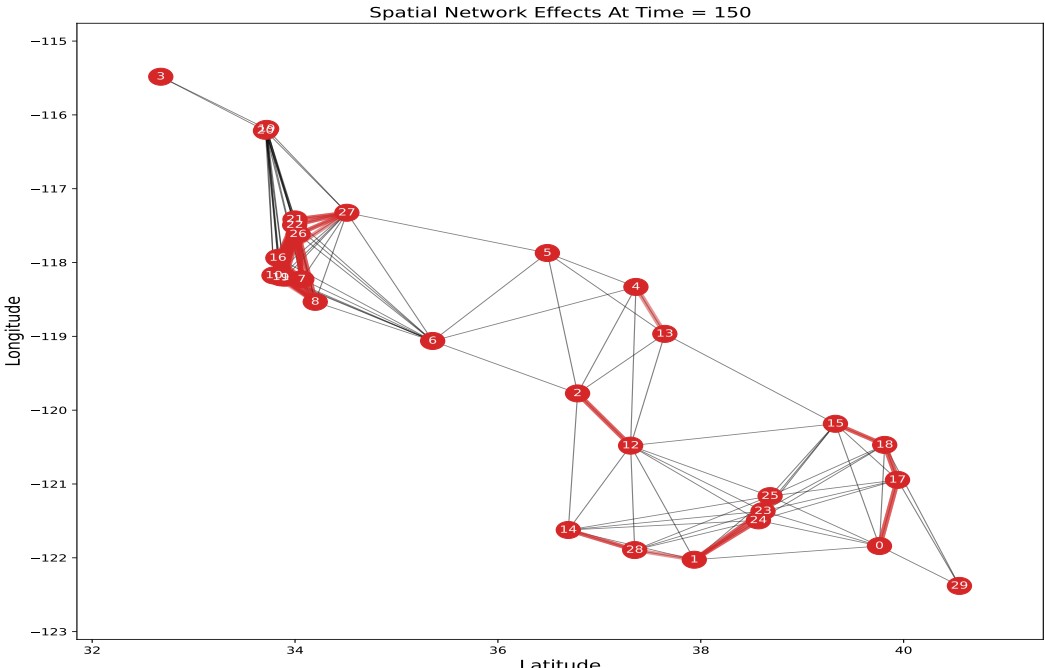

Figure 9: The distance-based effect among all 30 locations at time $t = 150$. Red lines show the connection larger than 75% of maximum value of estimated shape functions while gray lines are those between 50% and 75%.

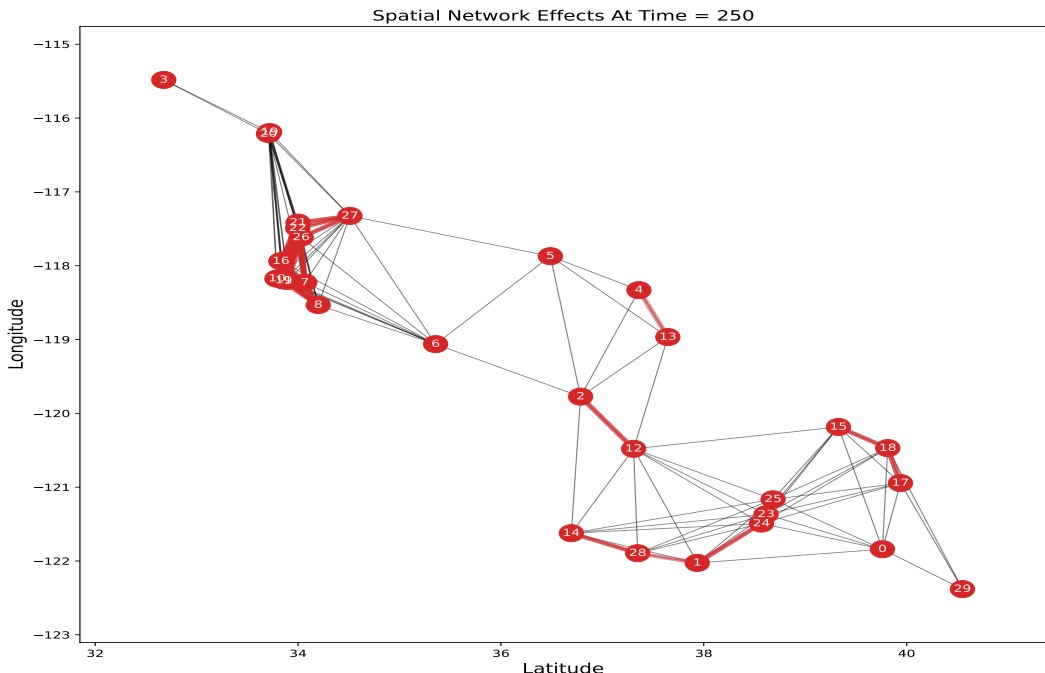

Figure 10: The distance-based effect among all 30 locations at time $t = 250$. Red lines show the connection larger than 75% of maximum value of estimated shape functions while gray lines are those between 50% and 75%.

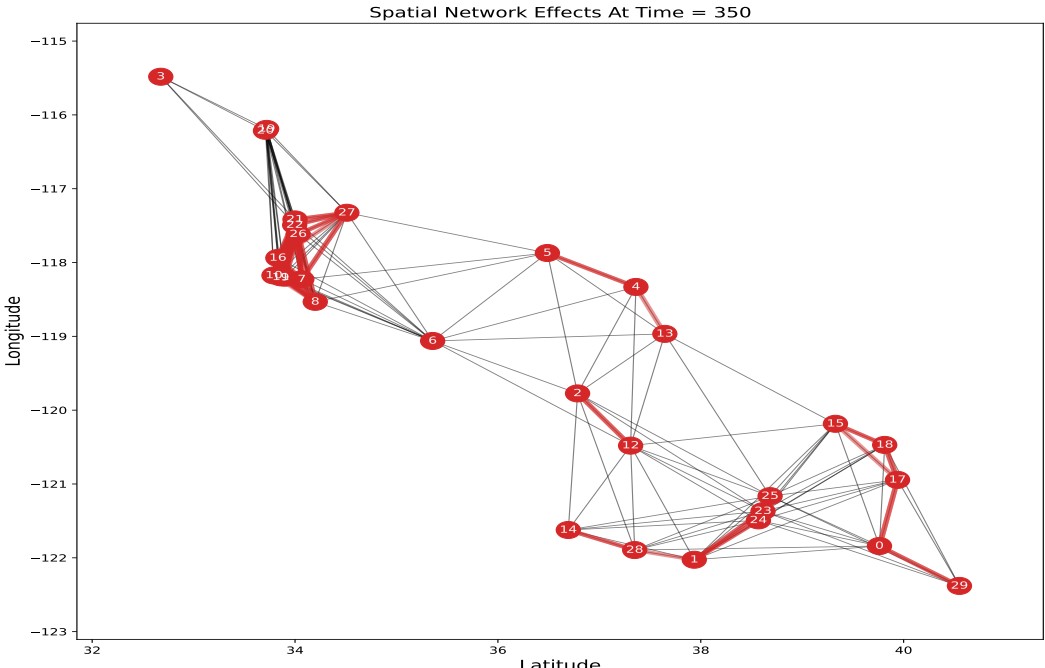

Figure 11: The distance-based effect among all 30 locations at time $t = 350$. Red lines show the connection larger than 75% of maximum value of estimated shape functions while gray lines are those between 50% and 75%.

 **C.2** $SO_2$ **data**

 The spatial information where the $SO_2$ data collected is list in Table 3. From the table we can obtain
 the name and coordinates for each site.

Table 3: The sites information for $SO_2$ data at 31 locations.

| Data Index | Site Name | County Code | State Code | latitude | longitude |
|---|---|---|---|---|---|
| 0 | Calaveras Lake | 29 | 48 | 29.275381 | -98.311692 |
| 1 | San Antonio Gardner Road | 29 | 48 | 29.352911 | -98.332814 |
| 2 | Dallas Hinton | 113 | 48 | 32.820061 | -96.860117 |
| 3 | Midlothian OFW | 139 | 48 | 32.482083 | -97.026899 |
| 4 | El Paso Chamizal | 141 | 48 | 31.765685 | -106.455227 |
| 5 | Fairfield FM 2570 Ward Ranch | 161 | 48 | 31.797813 | -96.1031 |
| 6 | Texas City Ball Park | 167 | 48 | 29.385234 | -94.93152 |
| 7 | Longview | 183 | 48 | 34.06659 | -118.22688 |
| 8 | Houston-The Woodlands-Sugar Land, TX | 201 | 48 | 29.623889 | -95.474167 |
| 9 | Park Place | 201 | 48 | 29.686389 | -95.294722 |
| 10 | Clinton | 201 | 48 | 29.733726 | -95.257593 |
| 11 | Houston Deer Park 2 | 201 | 48 | 29.670025 | -95.128508 |
| 12 | Hallsville Red Oak Road | 203 | 48 | 32.470228 | -94.481595 |
| 13 | Big Spring Midway | 227 | 48 | 32.280422 | -101.407137 |
| 14 | Borger FM 1559 | 233 | 48 | 35.6762 | -101.4401 |
| 15 | Beaumont Downtown | 245 | 48 | 30.036422 | -94.071061 |
| 16 | Port Arthur West | 245 | 48 | 29.897516 | -93.991084 |
| 17 | Port Arthur West 7th Street Gate 2 | 245 | 48 | 29.8442 | -93.9652 |
| 18 | Kaufman | 257 | 48 | 32.564968 | -96.317687 |
| 19 | Waco Mazanec | 309 | 48 | 31.653086 | -97.070704 |
| 20 | Corsicana Airport | 349 | 48 | 32.031934 | -96.399141 |
| 21 | Richland Southeast 1220 Road | 349 | 48 | 31.9041 | -96.352 |
| 22 | Corpus Christi West | 355 | 48 | 27.76534 | -97.434262 |
| 23 | Corpus Christi Tuloso | 355 | 48 | 27.832413 | -97.555387 |
| 24 | Corpus Christi Huisache | 355 | 48 | 27.804489 | -97.431553 |
| 25 | Orange 1st Street | 361 | 48 | 30.153675 | -93.725897 |
| 26 | Amarillo 24th Avenue | 375 | 48 | 35.236736 | -101.787405 |
| 27 | Amarillo Xcel El Rancho | 375 | 48 | 35.3165 | -101.7418 |
| 28 | Franklin Oak Grove | 395 | 48 | 31.168889 | -96.481944 |
| 29 | Tatum CR 2181d Martin Creek Lake | 401 | 48 | 32.277929 | -94.570851 |
| 30 | Cookville FM 4855 | 449 | 48 | 33.0752 | -94.8474 |
| 31 | Austin North Hills Drive | 453 | 48 | 30.354944 | -97.761803 |

 We obtain the first 200 steps for training and perform forecasting for the next 165 steps. All models
 are trained for 100 epochs using Adam optimizer at a learning rate of 0.01 and batch size of 50. The
 forecasting performance in Figure 12 and Figure 13. Figure 14 show the estimated shape functions.
 The dynamics of distance-based effects are presented in Figure 15, Figure16 and Figure 17.

 The Houston cluster is centered at the location with latitude 30 and longitude -94, while the Dallas
 cluster gathers around the site with latitude 32 and longitude -95. The results show that there exist
 interaction within each cluster. Interestingly we can see there are also some connection between the
 two main cluster at some times. The most strongest that larger than given threshold is denoted in red,
 which represent the significant distance-based effect at given time.

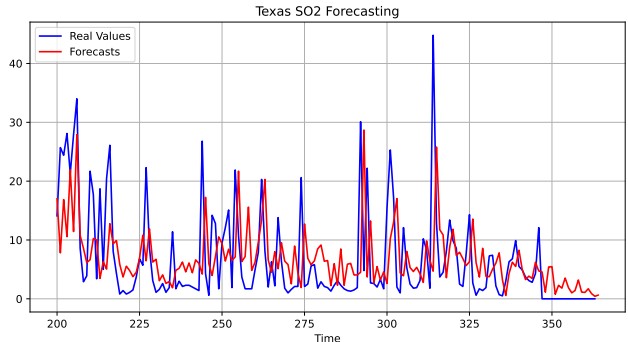

Figure 12: The sample of forecasting of $SO_2$ in one location for the next 165 days.

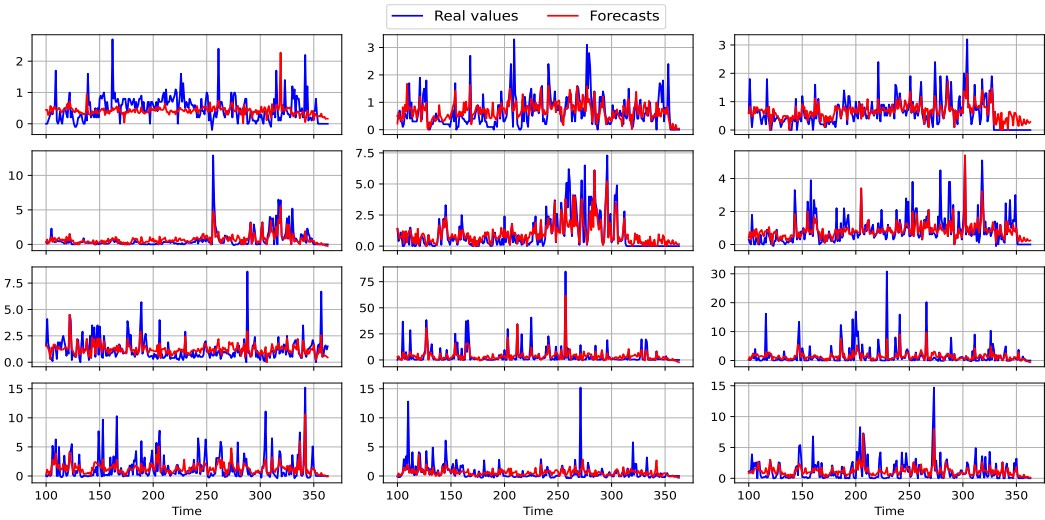

Figure 13: The sample of forecasts for other 12 locations.

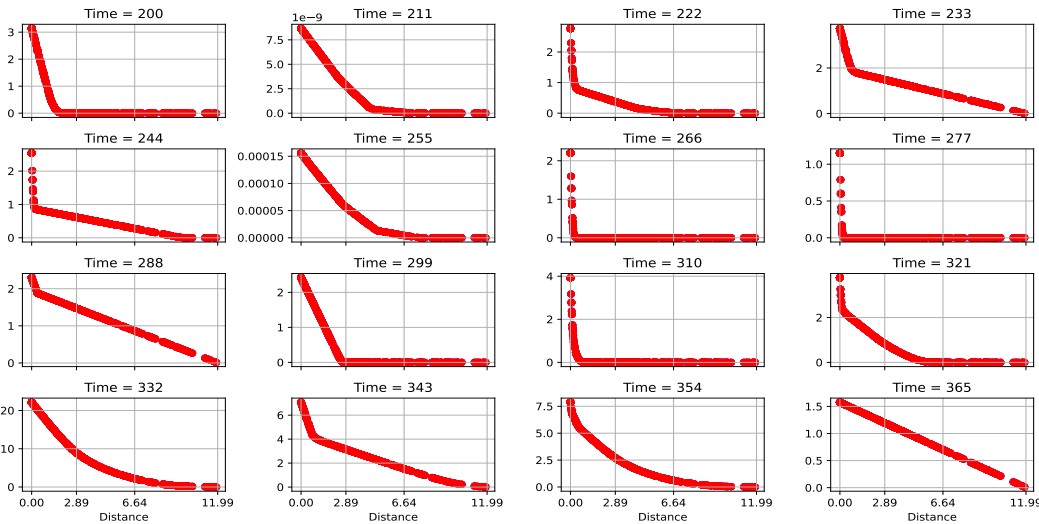

Figure 14: The samples of estimated shape function within the next 165 testing time steps. Distances are shown every $40^{th}$ quantile.

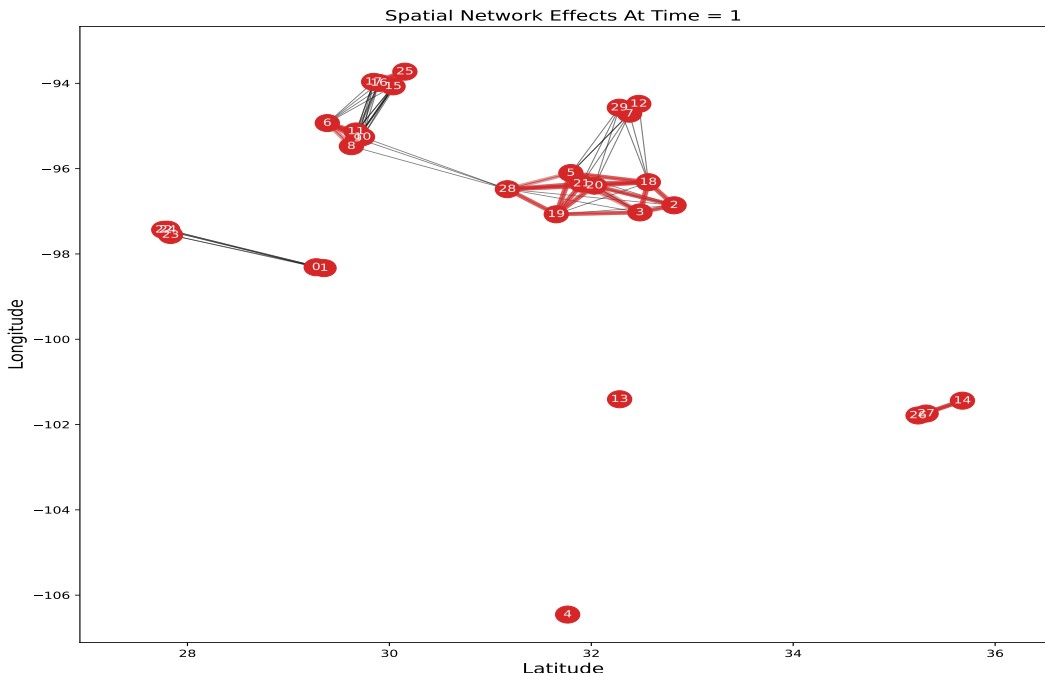

Figure 15: The distance-based effect among all 31 locations at time $t = 1$. Red lines show the connection larger than 75% of maximum value of estimated shape functions while gray lines are those between 50% and 75%.

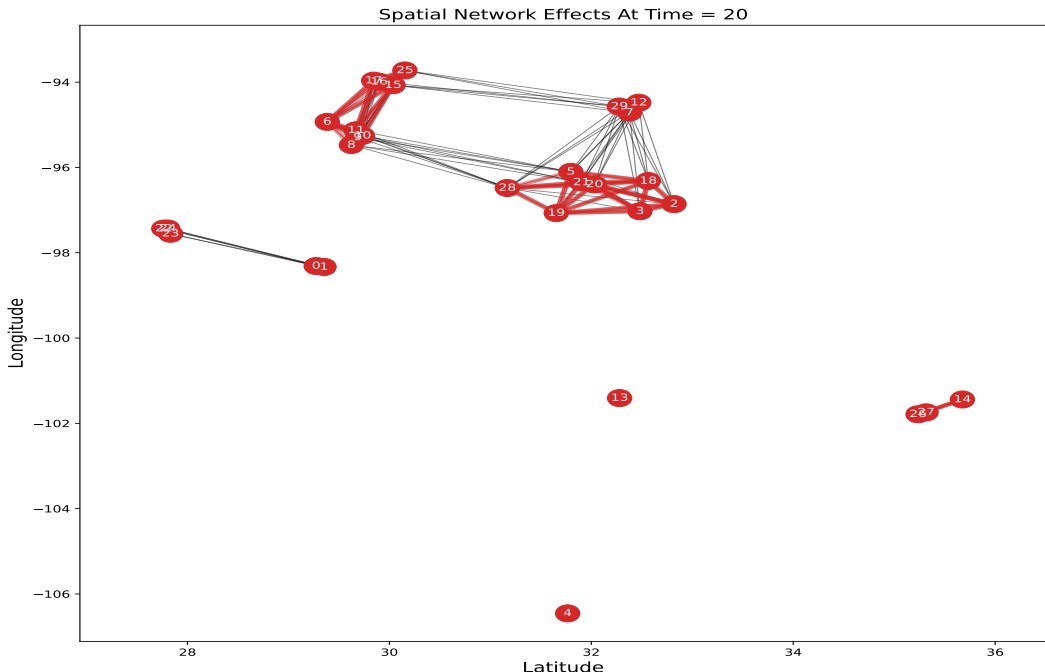

Figure 16: The distance-based effect among all 31 locations at time $t = 20$. Red lines show the connection larger than 75% of maximum value of estimated shape functions while gray lines are those between 50% and 75%.

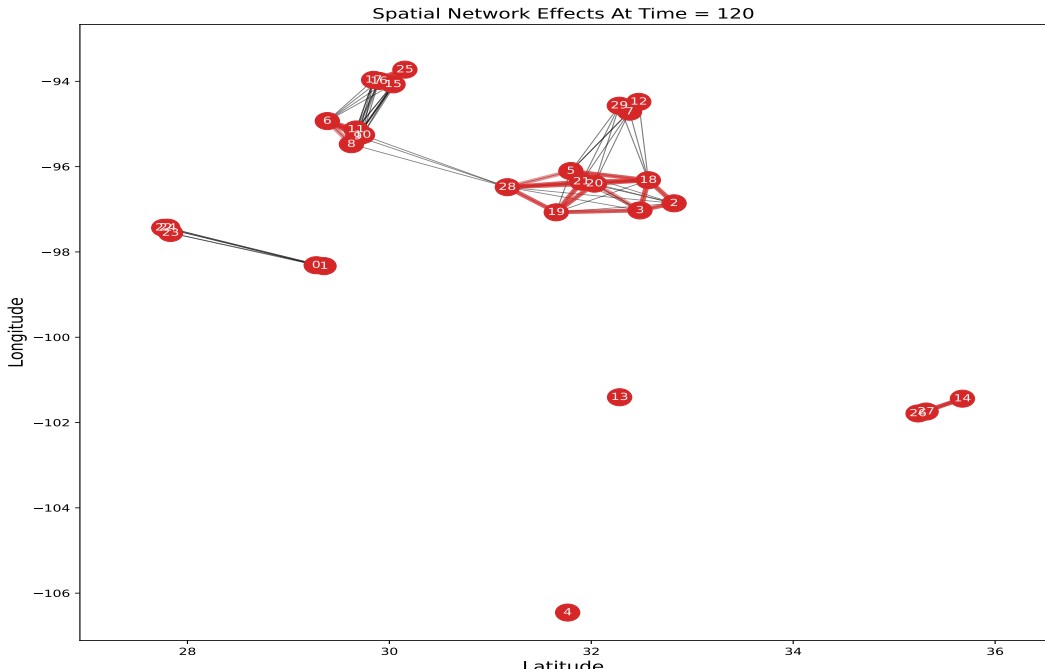

Figure 17: The distance-based effect among all 31 locations at time $t = 120$. Red lines show the connection larger than 75% of maximum value of estimated shape functions while gray lines are those between 50% and 75%.