# OpenReview forum: "Explainable Spatio-Temporal Forecasting with Shape Functions"
_NeurIPS.cc/2022/Conference — NeurIPS 2022 Submitted_

### Official Review · Reviewer_tdLh · 2022-07-11

**Rating:** 4
**Confidence:** 4
**Soundness:** 3 good
**Presentation:** 3 good
**Contribution:** 2 fair

**Summary:**

The paper proposes a spatio-temporal prediction model that tries to incorporate a varying strength of spatial dependency over distance. The idea is to model spatial dependency weights as a shape constraint. There are some empirical evaluations on simulation datasets and real-world datasets.

**Questions:**

1. The Gaussian process model is geostatistics also captures the effect of weakening spatial dependency with increasing distance. How does the proposed idea compare with the Gaussian process strategy?

2. The proposed idea is under the category of traditional machine learning methods. It is suspicious that the proposed model can outperform deep learning models in prediction accuracy. How do you explain the results in Table 1?

**Limitations:**

The paper does have some interesting discussions on the limitations of the proposed idea, e.g., not capturing the non-Euclidean space like spatial networks, and the lack of causal interpretations.

**Strengths And Weaknesses:**

Strength:

* The paper solves an important problem, i.e., training explainable spatio-temporal forecasting model that can automatically learn the dependency structure.

* The paper is overall well-written and easily understandable.

Weakness:

* Although the problem of learning spatio-temporal dependency structure is interesting, the proposed solution is not well-justified. The proposed method explicitly models spatial dependency as a function of distance. But the similar philosophy has already been explored in spatial statistics, such as the Gaussian process (it represents the covariance matrix of different sample locations based on distance between those locations).

* The proposed model seems limited in model representation with a linear function. Currently, it is not convincing that such models can achieve better performance than recent deep learning models.

---

> ### Author Response · Authors · 2022-08-02
> **Response to the reviewer tdLh**
>
> Thanks for your work and thoughtful comments on the work. We appreciate that you confirm the importance of the research topic and acknowledge our work's presentation. We also understand your concerns and we reply to each of them below.
>
> **Q: Although the problem of learning spatio-temporal dependency structure is interesting, the proposed solution is not well-justified. The proposed method explicitly models spatial dependency as a function of distance. But the similar philosophy has already been explored in spatial statistics, such as the Gaussian process (it represents the covariance matrix of different sample locations based on distance between those locations). The Gaussian process model in geostatistics also captures the effect of weakening spatial dependency with increasing distance. How does the proposed idea compare with the Gaussian process strategy?**
>
> A: We notice that a similar philosophy has already been explored in spatial statistics. Besides the Gaussian process, spatial regression and panel models are intensively developed in the statistical community. However, as we introduced in the manuscript, these statistical models have disadvantages or limitations. Besides, most of these statistical models focus on estimating single model parameters, which restrict them from being widely applied in real case study. Furthermore, such settings that spatial weight matrix specified on distance are fixed or subjective, not learnable. We also justified the proposed ESTF model with classical statistical models in experiments. In summary, restrictions of statistical models and non-learnable parameters motivate us to propose the ESTF model.
>
> In geostatistics and the Gaussian process, the covariance function is often assumed to be parametric, such as the Mat\'{e}rn class. The key difference of the proposed method is a nonparametric method and the only requirement is that the interaction between two locations decreases as the distance increases.
>
> **Q: The proposed model seems limited in model representation with a linear function. Currently, it is not convincing that such models can achieve better performance than recent deep learning models. The proposed idea is under the category of traditional machine learning methods. It is suspicious that the proposed model can outperform deep learning models in prediction accuracy. How do you explain the results in Table 1?**
>
> A: The proposed idea is not necessarily better but still comparable and more explainable. In our experiment, DC-RNN and our ESTF method can achieve similar results. For other baseline models, the forecasting accuracy is comparable under some scenarios. It is worthy noting that the proposed model consumes less training time than deep learning models. Furthermore, it enhances the interpretability compared with most of DNNs.

---

> ### Author Response · Authors · 2022-08-06
> **A friendly reminder**
>
> Dear Reviewer tdLh,
>
> We appreciate your comments and time! We have provided answers to your questions and revised the paper following your suggestions. Would you mind checking it and confirming if you have further questions?
>
> Best Regards,

---

> ### Author Response · Authors · 2022-08-08
> **A friendly reminder**
>
> Dear Reviewer tdLH,
>
> We appreciate for your work again and we have made some responses for what you concern. As the discussion deadline is approaching, could you please tell us any further concerns you have and we can make clarification accordingly.
>
> We believe discussions can make our work solid and your comments can drive our research to a high level. Could you please let us know if you have any further questions?
>
> Regards,

---

### Official Review · Reviewer_JW2N · 2022-07-12

**Rating:** 7
**Confidence:** 3
**Soundness:** 3 good
**Presentation:** 3 good
**Contribution:** 3 good

**Summary:**

This manuscript focuses on the patio-temporal forecasting problem and proposes a new method with shape functions toward a learnable and explainable model. The proposed method extends the statistical models via learnable bash shape functions, making it possible to capture the spatial variability and distance-based effects over distance. The proposed method is evaluated on both real and synthetic datasets.

**Questions:**

- It would be interesting to add more discussion about complex shape functions.

- For the evaluation, it is also suggested to add the inference time beyond the training time, which is more important in real scenarios.


**Limitations:**

Yes

**Strengths And Weaknesses:**

Pros:

- The idea of modeling the spatial weight matrix with learnable shape functions is novel and interesting. It generalizes the traditional statistical method by introducing more learnable components.

- Experimental results on both synthetic and real datasets indeed show the effectiveness of the proposed method. It even performs better or is comparable with respect to some DNN-based methods under some scenarios.

Cons:

- The proposed method is only evaluated on a single real-world dataset, making it hard to figure out if the method can consistently work well under real setups.

- The authors provide several specifications of the relations between the distance and the weights. However, the authors did not discuss the advantage or disadvantages of each specification. On a given dataset, how should we choose among them?

- According to section 3.4, the basis function for the shape constraints is simple. There is no discussion about some more complex functions and their influences on the performances.

Minor issues:

- "modelling" should be "modeling"

- The citation format is a little strange with semicolons and parentheses, it is better to reformat them.

- For better visualization, it is highly recommended to adopt vector graphics for all figures.

---

> ### Author Response · Authors · 2022-08-02
> **Response to the reviewer JW2N**
>
> Thanks for your work and valuable comments on our model. We are happy to hear from you that you acknowledge our work, particularly in the following aspects:
>
> 1. **The idea is novel and interesting.**
>
> 2. **Traditional statistical methods are generalized by learnable components.**
>
> 3. **Its performance is comparable with some DNN methods and even better in some cases.**
>
> We treasure your constructive comments on the weakness of the work. Each of them is answered below. We are welcome to further discussion to make our work better.
>
> **Q: The proposed method is only evaluated on a single real-world dataset, making it hard to figure out if the method can consistently work well under real setups. For the evaluation, it is also suggested to add the inference time beyond the training time, which is more important in real scenarios.**
>
> We add another real case study and inference time in the manuscript. The real case study predicts $SO_2$ in Texas in 2021 and the result is consistent with the air quality data set used in the previous version. We update our manuscript and supplemental file to present these results. In $SO_2$ data, the two significant clusters represent counties around Houston and Dallas, showing the strong connection between these locations and how distance-based effects interacted. The forecasting procedures are also conducted and we  list numerical results in Table 2. In both datasets, our ESTF model performs best under the RMSE metric, while DC-RNN is best in the MAE metric. In terms of training time, the ESTF model can reduce three or four times time consumption compared with baseline models. We also provide inference time as the reviewer suggested, which is comparable between the proposed method and the deep learning methods.
>
> ### Table 1 The error metrics with baseline models for Air quality data
> | Models       | MAE           | RMSE  |  Training Time (s) | Inference Time (s)|
> | ------------- |:-------------:| -----:|------------|---------------|
> | VAR     |    16.9844    | 22.3410         | 3.56        | 0.04       |
> | SPM   |   8.4547    |  13.8262      | 0.31       | 0.03    |
> | DC-RNN |  **4.7157**     | 9.3873     | 203         |**1.211**      |
> |FC-GAGA| 7.8671 |18.1870 |181 |2.759 |
> |GMAN|12.5268 |17.3817|140 |1.823 |
> |ConvLSTM|12.6292 |17.9149 |53 |1.940 |
> |**ESTF**| 5.2237|**9.2169** |**22** |1.625 |
>
>
> ### Table 2 The error metrics with baseline models for $SO_2$
> | Models       | MAE           | RMSE  |  Training Time (s) | Inference Time (s)|
> | ------------- |:-------------:| -----:|------------|---------------|
> | VAR     |    6.2705   | 9.1388    | 3.330   | 0.016     |
> | SPM   |   7.1453 | 9.1086   |0.143       |0.027    |
> | DC-RNN |  **3.5094**     | 6.8681     | 264.215     |**1.366**      |
> |FC-GAGA| 4.5976 |7.7528 |169.425 |2.889 |
> |GMAN|4.1099|7.4806|172.016 |1.581 |
> |ConvLSTM|4.1445 |8.0688 |96.233 |1.656 |
> |**ESTF**| 4.2966|**6.8307** |**31.050** |1.868 |
>
> **Q: The authors provide several specifications of the relations between the distance and the weights. However, the authors did not discuss the advantage or disadvantages of each specification. On a given dataset, how should we choose among them?**
>
> A: In the previous version, we provided several specifications for the spatial weight matrix. These specifications originate from the spatial model, in particular, the spatial autoregressive models and spatial panel models. In spatial statistics/econometrics, one main reason for the normalization (that is, the displayed three specification in Section 3.2 of the previous version) is to ensure the eigenvalue of spatial weight matrix is less than $1$, and thus, to avoid explosive value (that is,  $\bf{X}_t \rightarrow \infty$ as $t \rightarrow \infty$.)
>
> After reading the feedback of reviewers, we realize that other specifications are not needed. First, all other specifications leads to a particular type of the decreasing function, and therefore, it is best to directly specify the spatial weight as a decreasing function. Moreover, our simulation study and data analysis shows that the additional normalization is not needed since the estimated spatial weight matrix does not lead to explosive value.
> Therefore, we have removed other specifications.

---

> > ### Author Response · Authors · 2022-08-02
> > **Response to the reviewer JW2N**
> >
> > **Q: According to section 3.4, the basis function for the shape constraints is simple. There is no discussion about some more complex functions and their influences on the performances. It would be interesting to add more discussion about complex shape functions.**
> >
> > A: Although basis functions with shape constraints are simple, such monotone, concave or convex decreasing, the estimated shape functions can be uniquely determined by distance samples and keep consistency for real shape function [1]. Although shape constraints are simple, the estimated function that combine these basis can approximate the real shape function as the number of sample data increase. The theory stated in [1] proof that such basis functions can consistently approximate the continuous function that also satisfies shape constraints. The monotone decrease constraint is common under the context of spatio-temporal modeling and it can reflect the main feature of spatial dependence across distance.
> >
> > [1]. Chen, Y. and Samworth, R.J., 2016. Generalized additive and index models with shape constraints. Journal of the Royal Statistical Society: Series B (Statistical Methodology), 78(4), pp.729-754.
> >
> > **Q: Some minor issues on words, citation and format of figures.**
> >
> >  A: Thanks for your careful check on the presentation of our work. We update the manuscript accordingly.
> >
> > **Q: It would be interesting to add more discussion about complex shape functions.**
> >
> > A: The complex shape functions depend on real-world datasets, and  are less common than the decreasing shape function.  For example, if a system is accelerating (that is, the second-order  of $\bf{X}$ is positive), a convex constraint may be needed. Due to the page limit, we do incorporate it in the main manuscript.
> >
> > **Q: For the evaluation, it is also suggested to add the inference time beyond the training time, which is more important in real scenarios.**
> >
> > A: We have added the inference time to the real case studies.

---

> > ### Author Response · Authors · 2022-08-06
> > **A friendly reminder**
> >
> > Dear Reviewer JW2N,
> >
> > We appreciate your comments and time! We have provided answers to your questions and revised the paper following your suggestions. Would you mind checking it and confirming if you have further questions?
> >
> > Best Regards,

---

### Official Review · Reviewer_WyxW · 2022-07-12

**Rating:** 6
**Confidence:** 4
**Soundness:** 3 good
**Presentation:** 3 good
**Contribution:** 3 good

**Summary:**

In this paper, the authors propose an interpretable spatio-temporal forecasting method by learning shape functions from data. The shape function is designed as a function of the distance between pairwise locations, which is then used to represent the vector autoregressive model coefficients. As a result, the proposed model can incorporate both spatial and temporal information. Furthermore, the interactions between different locations can be interpreted by the shape function outputs, and a graph can be used to visualize the interactions. Finally, the proposed method also achieves promising forecasting performance compared to deep learning methods.

**Questions:**

Questions:
In Eq (1), the authors only consider time lagged correlations. Is it possible to incorporate instantaneous relations at the same time step? The reason why I ask this question is because there might exist instantaneous relations in some applications.

What is the motivation behind the choice of specifications in section 3.2? Are the authors trying to mimic some properties in the classical statistical methods?

**Limitations:**

The authors addressed the limitations.

**Strengths And Weaknesses:**

Strength:
The work is well motivated by the challenge in spatio-temporal forecasting, i.e., the tradeoff between accuracy and interpretability. The proposed method uses shape functions to constrain the statistical relations between different locations and achieves good forecasting performance. Overall, I think the paper makes a good contribution to spatio-temporal forecasting.

The explanation of relations to previous works is clear. The proposed method is an extension of traditional methods by replacing the predefined spatial functions with learnable shape functions. The proposed method enjoys good forecasting accuracy as deep learning methods, while maintaining the interpretability of traditional methods.

Weakness:
The main assumption of this paper is the correlation between two locations decreasing with distance. While this assumption may make sense in some applications, it would be better to give some discussions or case studies of this assumption.

In the nonstationary setting, the model in Eq (2) has a different weight matrix for each time step. It seems to be that this design suffers from the small sample problem, as the data at each time step is limited. This would cause fluctuations in the estimations across time.

In section 3.4, the authors give examples of monotonically increasing/deceasing functions. However, in the experiments, only monotonically decreasing functions are investigated. If increasing functions are not going to be used in the forecast model, it might be better not to mention it the method section.

---

> ### Author Response · Authors · 2022-08-02
> **Response to the reviewers WyxW**
>
> Thanks for your work and comments. We are happy to see that the reviewer acknowledges our work solving challenges in spatio-temporal forecast,  particularly:
>
> 1. **Contribute current works with the same topic.**
>
> 2. **Solve the challenge of the trade-off between accuracy and interpretability.**
>
> 3. **The explanation of relations to previous works is clear.**
>
>  We also understand your concerns on weakness or some questions you pointed out. We reply to them below:
>
> **Q: The main assumption of this paper is the correlation between two locations decreasing with distance. While this assumption may make sense in some applications, it would be better to give some discussions or case studies of this assumption.**
>
> A: Our model is related to spatial statistics, and some ideas originated from the statistical community. In the spatial statistical community, the assumption is known as Tobler's First Law, which is "Everything is related to everything else, but near things are more related than distant things"[1][2]. Such assumptions are well-accepted by researchers and widely adopted in some research communities. Many scenarios in natural phenomena, such as geology, atmospheric science, etc., follow the rule based on distance. Take the example in the manuscript. The air quality shows such characteristics across distances. We also add another real case study to verify the point.
>
> [1] Tobler, W.R., 1970. A computer movie simulating urban growth in the Detroit region. Economic geography, 46(sup1), pp.234-240.
>
> [2] Miller, H.J., 2004. Tobler's first law and spatial analysis. Annals of the association of American geographers, 94(2), pp.284-289.
>
> **Q: In the non-stationary setting, the model in Eq (2) has a different weight matrix for each time step. It seems to be that this design suffers from the small sample problem, as the data at each time step is limited. This would cause fluctuations in the estimations across time.**
>
> A: We also notice this point in practice. We can provide several justification and improvement measures on the issue. Firstly, the smaller sample problem can be solved by increasing the number of locations. It should be noted that the input data is the pairs of observational differences across distance instead of observation directly. For example, observation at 100 locations can generate around 5000 input data.
>
> The fluctuations during the estimation procedures can be solved by adding smoothing techniques. The smoothing techniques is to increase training windows instead of only consider each time step. For instance, when estimate data at time $t$, we can use data in the time windows from $t-5$ to $t+5$.
>
> **Q: In section 3.4, the authors give examples of monotonically increasing/deceasing functions. However, in the experiments, only monotonically decreasing functions are investigated. If increasing functions are not going to be used in the forecast model, it might be better not to mention it the method section.**
>
> A: Thanks for your suggestion. We agree that the increasing functions are not needed here, and has removed it.

---

> > ### Author Response · Authors · 2022-08-02
> > **Response to the reviewers WyxW**
> >
> > **Q: In Eq (1), the authors only consider time lagged correlations. Is it possible to incorporate instantaneous relations at the same time step? The reason why I ask this question is because there might exist instantaneous relations in some applications. What is the motivation behind the choice of specifications in section 3.2? Are the authors trying to mimic some properties in the classical statistical methods?**
> >
> > A: In some applications, some research takes the instantaneous effect into account. If $X_t$ represents spatio-temporal process, the instantaneous relations are referred to as pure spatial dependence in some literature, as only observation simultaneously is used to estimate parameters. The classical spatial statistical models considering the setting can be found in some spatial autoregressive models or spatial panel models.
> >
> > In this work, the proposed model aims to make forecasts in spatio-temporal scenarios. Therefore, it is critical to prioritize time-lagged correlations and model relationships among time lags. On the one hand, many time-series-based models only model relationships between different time lag when making forecasts. On the other hand, incorporating instantaneous effects simultaneously may bring difficulties when making forecasts. For example, given the training data with size $T$, the forecast for $X_{T+1}$ may depend on $X_{T+1}$. In practice, integrating it is unnecessary as no one knows the exact value at $T+1$.
> >
> >
> > For specifications in section 3.2 and the motivation behind it. First, it should be noted that some specifications for spatial weight matrix are considered under the context of increasing functions. As we stated in the last questions, we take the inverse of increasing functions to form the spatial weight matrix, making elements of the matrix reflect distance-varying characteristics. Some classical statistical models inspire the other point motivating us to select such specifications. For ease of interpretation, it is common practice to normalize $W$ such that the elements of each row sum to unity. Since $W$ is non-negative, this ensures that all weights are between 0 and 1, and has the effect that the weighting operation can be interpreted as an averaging of neighboring values [3]. Besides, estimation procedures require that the eigenvalue of $W$ is less than 1 to satisfy some properties in the spatial statistical community.
> >
> > After reading the feedback of reviewers, we realize that other specifications are not needed. First, all other specifications leads to a particular type of the decreasing function, and therefore, it is best to directly specify the spatial weight as a decreasing function. Moreover, our simulation study and data analysis shows that the additional normalization is not needed since the estimated spatial weight matrix does not lead to explosive value.
> > Therefore, we have removed other specifications.
> >
> > [3] Elhorst, J.P., 2014. Spatial econometrics from cross-sectional data to spatial panels. Springer.

---

> > ### Author Response · Authors · 2022-08-06
> > **A friendly reminder**
> >
> > Dear Reviewer WyxW,
> >
> > We appreciate your comments and time! We have provided answers to your questions and revised the paper following your suggestions. Would you mind checking it and confirming if you have further questions?
> >
> > Best Regards,

---

### Author Response · Authors · 2022-08-02
**General response to reviewers**

We appreciate all reviewers' work and friendly comments. Your valuable suggestion can help us to improve the quality of the work and encourage us to pursue perfection in research.

All reviewers acknowledge that our work aims to solve an important problem, forecast in spatio-temporal scenarios. It generalizes traditional statistical models with learnable components such as making spatial connections explainable. The results are comparable with respect to some DNN models and outperform them in some scenarios. We appreciate reviewers confirming our work novelty and contributing to current spatio-temporal forecast problems. The modification in the manuscript is summarized as follows:

1.**We add another case study to verify consistency under real setups.**

2.**We provide inference time beyond the training time.**

3.**We modify some introduction and discussion to response reviewers' comments. Some minor issues are revised accordingly.**

We also appreciate reviewers pointing out our weaknesses. We reply to them for each reviewer and update our manuscript accordingly. The modification parts are marked in blue colour.

We believe your constructive suggestions and comments can make the work more solid and perfect.

---

### Meta-Review · Area_Chair_HJat · 2022-08-27

**Recommendation:** Reject
**Confidence:** Certain

**Metareview:**

The paper proposes to use shape functions as basis functions to characterize spatial dependencies. They incorporate the shape functions into the spatial regression model and demonstrate strong forecasting performances compared to graph convolution-based methods. While the techniques are interesting, it is not directly relevant to the ICLR (deep learning) community. The argument for explainability is also subjective and less convincing.

**Award:**

No

---

### Decision · Program_Chairs · 2022-09-14

Reject